# Prostate-Specific Antigen and Testosterone Levels as Biochemical Indicators of Cognitive Function in Prostate Cancer Survivors and the Role of Diabetes

**DOI:** 10.3390/jcm10225307

**Published:** 2021-11-15

**Authors:** Alicja Popiołek, Bartosz Brzoszczyk, Piotr Jarzemski, Aleksandra Chyrek-Tomaszewska, Radosław Wieczór, Alina Borkowska, Maciej Bieliński

**Affiliations:** 1Department of Clinical Neuropsychology, Collegium Medicum in Bydgoszcz, Nicolaus Copernicus University in Toruń, 85-067 Bydgoszcz, Poland; aleksandrachyrek@wp.pl (A.C.-T.); alab@cm.umk.pl (A.B.); bielinskim@gmail.com (M.B.); 2Department of Internal Diseases, Jan Biziel University Hospital No. 2 in Bydgoszcz, 85-163 Bydgoszcz, Poland; wieczorcmumk@tlen.pl; 3Department of Laparoscopic, General, and Oncological Urology, Jan Biziel University Hospital No. 2 in Bydgoszcz, 85-067 Bydgoszcz, Poland; bartosz.brzoszczyk@gmail.com (B.B.); piotr@jarzemski.pl (P.J.)

**Keywords:** prostate cancer, cognitive function, cancer-related cognitive impairment, biomarkers, prostate-specific antigen, testosterone, diabetes

## Abstract

Prostate cancer (PC) is one of the most common malignancies in men. The increase in the number of PC survivors is associated with many problems including cognitive impairment. Early detection of such problems facilitates timely protective intervention. This study examined the association between prostate-specific antigen (PSA) or testosterone (T) levels and cognitive function in patients undergoing radical prostatectomy. Such a correlation could help identify patient groups at risk of cognitive impairment. Participants underwent clinical (demographic data, medical history, physical examination, and blood analyses) and neuropsychological assessment (cognitive test battery). Preoperative PSA or T levels were not associated with cognitive function. However, long-term follow-up after prostatectomy showed a strong correlation between PSA levels and the results of verbal memory and executive function tests. A trend toward significance was also observed for visuospatial memory. The levels of free T and total T were not correlated with cognitive function. Only the levels of free T after hormonal treatment were significantly correlated with executive functions. Comorbid diabetes affected these correlations. In conclusion, PSA levels at a distant postoperative time and free T level after hormonal treatment may be biomarkers of cognitive function.

## 1. Introduction

Prostate cancer (PC) is one of the most common malignancies in men [1]. Despite considerable progress in the detection and treatment of PC, it remains the fifth leading cause of death worldwide, with approximately 375,000 deaths per year. PC patients die more often from any cause than the general population [2,3]. Advances in medicine, particularly in screening and in the design of effective treatments, have led to an increase in the population of patients with PC, which requires specific care tailored to each problem. One of the issues that PC survivors have to face is cancer-related cognitive impairment. Deterioration of mental function occurs in patients with breast cancer, ovarian cancer, colorectal cancer, and lymphoma [4], as well as in PC [5,6,7]. It occurs in up to 75% of patients during cancer treatment and in up to 30% of patients prior to any treatment [8]. However, the pathogenetic mechanisms underlying these dysfunctions remain unclear. They seem to be multifactorial and related to both the cancer itself and to treatments such as radiochemotherapy, hormone therapy, and surgery [4,5,9,10]. The probable link of PC and cognitive dysfunction is supported by some biological processes including through abnormal accumulation of proteins, neuroinflammation, oxidative stress, and neuronal cell death. These changes can lead to the breakdown of the blood-brain barrier and promote key pathways for the development of cognitive disorders (such as brain insulin resistance, mitochondrial dysfunction, and deposits of neurotoxic beta-amyloid oligomers, synaptic loss, neuronal dysfunction, and cell death) [11,12,13].

Regardless of the mechanisms underlying their development, cognitive disorders affect the daily life of patients, decreasing quality of life, hindering cooperation, and interfering with the patient’s ability to live independently [4,8,14,15,16,17]. The comprehensive care of cancer survivors should include early assessment of cognitive function because early detection or even prediction of cognitive impairment enables timely protective intervention [16,18]. This may reduce the degree of cognitive loss and promote the maintenance of cognitive performance, which improves quality of life [19].

The development of cognitive disorders in cancer survivors is associated with age (the prevalence increases with age), genetic polymorphisms, psycho-social components, stress, depression, anxiety, or other mood disorders related to the disease [4,5,9,10].

This study examined the potential association of biochemical parameters that are routinely measured in PC, such as prostate-specific antigen (PSA) or testosterone, with cognitive function. The existence of such a dependency could help identify patients at risk of cognitive impairment in PC.

PSA is the biomarker recommended by clinical practice guidelines for the early detection of PC to rule-in patients for prostate biopsy referral, for PC surveillance, and for therapeutic monitoring [20]. Preliminary studies show an association between PSA level and cognitive decline [13,21]. Sternberg et al. suggested the possibility of using PSA as a serum biomarker of cognitive function [21].

Testosterone is a hormone that is routinely measured in PC. Its deficiency plays an important role in cognitive function impairment [22]. Androgens (testosterone among them) are closely related to beta amyloid. Their reduced levels induce beta amyloid accumulation in a brain and impair hippocampal neurons, causing cognitive difficulties [12].

Here, we evaluated the correlation of PSA and testosterone levels with cognitive function in PC survivors treated with radical prostatectomy and identified factors that could influence this correlation.

## 2. Materials and Methods

The study was carried out in The Department of Laparoscopic, General, and Oncological Urology of Jan Biziel University Hospital No. 2 in Bydgoszcz in Poland between July 2017 and June 2018. The study group comprised 118 men with PC who underwent laparoscopic radical prostatectomy. All patients were Polish and Caucasian. The mean age of the cohort was 65 years (range, 48–77 years). Patients were evaluated once postoperatively after an average of 26 months (range, 3–102 months). The inclusion criteria were as follows: histopathologically confirmed diagnosis of prostate adenocarcinoma and subsequent treatment with radical prostatectomy, the ability to understand the purpose of the study, and the willingness to participate in the study. The exclusion criteria were severe somatic, psychiatric, or neurological disorders.

All respondents underwent clinical and neuropsychological assessment.

Clinical assessment included personal data (age, ethnicity, education, physical activity, body mass index, and smoking status) and medical history (data about the course of the disease, PC Grade Group (GG) according to the International Society of Urological Pathology classification [23], date of prostatectomy, adjuvant therapy including radiotherapy and hormone therapy called androgen deprivation therapy (ADT), and comorbidities such as diabetes, hypertension, myocardial infarction, and stroke). Peripheral blood tests were also performed. The following parameters were determined: free and total testosterone levels and total PSA measured by enzyme immunoassay (the PSA score before prostatectomy, at week 6 after surgery, and at the time of the study, called current PSA, were used for data analysis), leukocyte levels, and C-reactive protein.

Neuropsychological assessment included assessment of cognitive function using the computer-based test Neurotest (Table 1) [24].

### Statistical Analysis

The distribution of the parameters of the study group was examined using the Shapiro–Wilk test and a significant deviation from the normal distribution was found. Statistical significance of differences between the groups was tested using the Mann–Whitney U test. The analysis of the significance of the correlation was tested using the R-Spearman correlation test. Analysis of covariance was used to investigate the effects of interactions (ANCOVA). The size of the effect is determined with Cohen’s d. A multiple regression model was used to perform multivariate analyses. Statistica 13.0 was used for statistical analysis.

## 3. Results

In the first stage of the analysis, the study group was classified according to a biochemical indicator of the effectiveness of the surgical procedure, namely a PSA level of <0.1 ng/mL. The study group was divided into those who achieved and those who did not achieve this result. Comparison of the clinicopathological and demographic characteristics of the two groups (Table 2) showed that the group with PSA > 0.1 ng/mL at 6 weeks postoperatively had a lower incidence of arterial hypertension and higher rates of disease severity before surgery according to GG classification. Analyses of biochemical parameters (Appendix A) showed that the group with a persistent increase in PSA levels had higher baseline PSA levels before therapy, as well as higher levels of PSA at the time of the study. Analysis of cognitive parameters (Appendix A) showed that this group also had a significantly lower number of remembered words in further attempts of the Verbal Memory (VM) test. There was also a trend toward longer response times in the Visuospatial Working Memory Task (VWMT).

The PSA levels before the procedure, at 6 weeks after the procedure, and the current levels were analyzed (Table 3). Preoperative PSA levels did not correlate with the results of cognitive tests. Postoperative PSA levels were correlated only with the lower score in the first attempt of the VM test. The current PSA level was positively correlated with a longer response time in the simple reaction test and with a lower number of remembered words in all VM tests and deferred memory trials. Longer response times in the GoNoGo test and VWMT were associated with higher PSA levels at the time of the study. There were no significant correlations between the level of total and free testosterone and the results of cognitive tests in the entire cohort (Appendix A).

The correlation between biochemical parameters and the results of cognitive tests was analyzed in subgroups of patients with arterial hypertension, stroke, myocardial infarction, and diabetes. Correlations were only detected in the subgroup of diabetic patients (*n* = 19), in which numerous and strong correlations between the current PSA level and the results of cognitive tests were observed (Table 4). After excluding diabetic patients, there was no correlation between the current PSA level and the results of cognitive tests in the remaining patients.

In the subgroup of diabetic patients, those with higher levels of free testosterone had a higher number of correct GoNoGo test responses and fewer incorrect Go responses in this test (Appendix A).

Because part of the study population underwent hormone therapy, we analyzed the correlation between biochemical parameters and cognitive function in subgroups of patients with and without hormone therapy. In patients who received hormone therapy (Table 5), the level of free testosterone was significantly associated with a greater number of remembered words in the VM deferred test, a greater number of correct answers in the GoNoGo test, and a lower number of incorrect Go and incorrect NoGo responses. Analysis of the correlation between current PSA levels and cognitive outcomes in the same groups showed that in both groups, a higher level of current PSA was associated with some cognitive parameters (Table 5).

To improve the accuracy of the analyses, we assessed whether the group of patients with diabetes differed from other patients (Appendix A). Patients with diabetes were characterized by significantly lower levels of physical activity and significantly lower levels of total testosterone. In the cognitive tests, diabetics had longer reaction times in the simple reaction test and made significantly more incorrect NoGo responses in the GoNoGo test (Appendix A).

A summary analysis of the covariance of factors determining the parameters of neuropsychological tests was also performed. This confirmed the significance of the studied biochemical and clinical factors, such as diabetes, in a cognitive context (Table 6). To confirm the obtained results, a multiple regression model was used, which showed a significant share of the current PSA level in almost all the results of cognitive tests. Other important factors were age, diabetes, and one-time post-surgery PSA (Appendix A).

## 4. Discussion

### 4.1. Prostate-Specific Antigen

PSA belongs to the serin protease family and together with other clinical tools (digital rectal examination, family history of PC, age, and magnetic resonance imaging) is recommended by clinical practice guidelines for the early detection of PC and for therapeutic monitoring [25]. In this study, PSA levels before prostatectomy did not correlate with cognitive function (Table 3). Postoperative PSA levels correlated only with the lower score in the first attempt of the VM test. However, when PSA levels were measured at longer times after prostatectomy, there was a significant relationship between PSA and cognitive functions including reaction time and VM (Table 3). This is consistent with the literature, and suggests that PSA could be used as a marker of cognitive function [21]. Sternberg et al. compared PSA levels between patients with Alzheimer’s disease (AD), mild cognitive impairment (MCI), and a control group. The results showed a significant association between cognitive status and PSA levels [21]. A similar study was performed by Lin et al. in a large cohort; these authors observed that patients with AD not only had increased PSA levels, but also an increased risk of PC compared with controls [13]. In studies that did not include PSA level, but only the diagnosis of PC and its treatment, neither PC itself nor the use of hormonal treatment was associated with MCI [26].

The above studies were limited to the overall assessment of cognitive function or the overall diagnosis of cognitive disorders, and did not assess individual domains. In the present study, PSA levels were associated with almost all cognitive domains, especially reaction time and VM. In addition, visuospatial memory was associated with PSA levels in the present cohort (Table 3).

The mechanisms underlying these associations are not clear. Given that PSA is present in cerebrospinal fluid (CSF), it can be hypothesized that PSA has a direct effect on brain tissue and brain function [27]. Significant differences in CSF PSA levels were observed between cancer and other prostate diseases [28]. However, in patients with PC, high CSF PSA levels are commonly associated with central nervous system cancer involvement/metastasis [29]. The clinical usefulness of these studies, however, is limited because CSF PSA values are usually too low [28].

The relationship between cognitive function and PSA could be related to the treatment; some studies have reported that cognitive function in PC depends on the type of treatment [30,31]. The role of ADT, which is often used as an adjuvant treatment, is emphasized. Consistently, the cognitive domains that correlate with PSA are the most frequently affected in PC patients receiving ADT [6]. Testosterone deprivation (under the influence of hormonal treatment) could affect both cognitive function and PSA levels [21,32]. However, patients with increased PSA levels after local treatment are at a high risk of subsequent progression and often undergo ADT, which causes a decrease in cognitive disorders [5,33,34]. In this study, the group of patients with persistently elevated PSA after surgery had worse VM than those with normal postoperative PSA levels (<0.1 ng/mL) (Appendix A). To exclude the effect of hormonal treatment on the relationship between cognitive presentation and PSA, we performed subgroup analyses. These analyses showed that strong correlations between these variables also occurred in patients who did not receive hormonal treatment, but they concerned other cognitive domains (Table 5).

This study also examined various factors that could modulate PSA levels and affect their correlation with cognitive function. Among the factors that could affect PSA or cognitive function, we analyzed arterial hypertension and history of stroke, myocardial infarction, or diabetes [35,36]. Most of these factors were not correlated with cognitive function except diabetes mellitus. The results are presented in Table 4. The differences concerned all the studied cognitive domains.

Diabetes is an important independent risk factor for cognitive impairment [37]. Elabbady et al. assessed the role of diabetes in prostate-related parameters and observed significant differences in the levels of PSA between patients with and without diabetes [38]. Similarly, Kobayashi et al. reported that, in men with higher blood glucose levels, PSA values were significantly reduced [36]. Diabetes mellitus is thus a factor that influences cognition and can affect PSA.

### 4.2. Testosterone

Testosterone level is an important factor that affects many aspects of PC. In this study, we investigated the association of serum total testosterone and free testosterone levels with cognitive function in PC survivors and did not find any significant relationships (Appendix A). However, literature data on the association of testosterone and its supplementation with cognitive function are not consistent. Some studies have reported significant dependencies [12,39,40], whereas other reports have not [41,42]. A review by Cai and Li demonstrated that low levels of free testosterone and total testosterone are significantly correlated with cognitive decline, as well as with an increased risk of AD in older men [12]. Pintana et al., on the other hand, have remarked that new reports highlight non-significant findings regarding the relationship between testosterone and cognitive function [41]. These incompatibilities can be explained by methodological differences as well as differences in age or ethnicity [41]. Placebo-controlled randomized clinical trials also showed different effects of testosterone supplementation on cognitive function [43].

The treatment of PC also affects testosterone levels. To examine the potential influence of hormonal treatment, we performed separate analyses of cognitive function in men who received ADT and those who did not. In patients receiving hormone therapy, we observed a significant correlation between free testosterone levels and cognitive function, and high testosterone levels were associated with better cognitive performance. These correlations were not observed in the group of people who did not receive ADT (Table 5). Data on the role of ADT in cognitive function are also inconsistent. A beneficial effect of ADT on cognitive status has been reported [44], whereas other studies do not support these observations or report subtle but significant adverse effects of ADT [45]. Andella et al. reviewed publications assessing the cognitive functioning of PC patients treated with ADT. Of the 31 studies included in the analysis, 16 studies found that this therapy had no adverse effects on cognitive function, while another 11 studies found that it negatively affected cognition. In four studies, the results were inconclusive. The authors also noted that future research should focus on further exploring brain features with functional magnetic resonance imaging because this technique may be more sensitive in detecting brain abnormalities in patients treated with ADT [46]. Neuroimaging studies indicate that ADT may cause changes in regional brain metabolism associated with alterations in spatial performance and VM. Brain areas that are affected by androgen deprivation are consistent with those affected in diabetes and in AD, suggesting possible common mechanisms [47]. These changes are located only in brain regions that control specific functions. This may explain why these abnormalities are often elusive in general cognitive function tests, as their detection requires methods aimed at assessing specific cognitive domains [48]. In studies of specific domains, patients treated with ADT perform worse than controls mostly in visuomotor tasks and spatial and VM [43,49,50]. In this study, we observed cognitive decline mostly in executive functions and deferred memory (Table 5).

As described for PSA, we performed additional analyses to identify factors affecting free testosterone levels and cognitive parameters. In patients with diabetes, the level of free testosterone strongly correlated with cognitive control and cognitive inhibition, and a correlation was also observed in the deferred memory test (Appendix A). Both diabetes and low testosterone levels are risk factors for cognitive dysfunction [51]. Bertram et al. reported that low testosterone level is a risk factor for both cognitive impairment and the development of diabetes. These findings suggest that the dependencies are complex. German scientists proposed a molecular framework that links diabetes, testosterone, and cognitive impairment [11]. They suggested that the blood-brain barrier can break down because of the inflammatory, oxidative, and metabolic changes in diabetes, which promote the pathological features of dementia. The relationship between testosterone levels, insulin-resistant obesity, and cognitive function was also observed by Pintana et al. [41]. This author proposed a model in which cognitive decline results from either the direct effect of lack of testosterone on the brain or an association between testosterone deficiency and obesity leading to fat tissue accumulation, insulin resistance, and finally cognitive impairment [41].

In conclusion, the present findings suggest that diabetes is associated with the effect of PSA or testosterone levels on cognitive function, supporting that good quality of care in diabetes patients with PC is important. However, the present study was limited by the small number of participants. Additional studies, including molecular studies, are needed to confirm and better understand the observed phenomena and their causes.

## 5. Conclusions

Persistently elevated PSA levels following prostatectomy and free testosterone levels after hormonal treatment are potential biomarkers of cognitive function that are related mainly to VM and executive function. Concomitant diabetes affects the correlation between PSA levels and cognitive abilities, as well as that between testosterone level and cognitive function.

## Figures and Tables

**Table 1 jcm-10-05307-t001:** The computer-based test battery (Neurotest) components.

Test	Patient’s Task	Evaluated Parameters
Simple reaction time test (SRT)	to press the button after seeing a green circle appearing on the computer screen; the stimulus is presented five times, and the number of correct answers and the average response time (ms) are measured	speed and correctness of reactions to stimuli; general vigilance and psychomotor speed
Verbal memory test (VM test)	to remember as many words as possible from the list of 10 words read by a researcher five times; the patient has to recall words in any order after each reading and 20 min later; the number of correctly repeated words, the number of words outside the list and the number of repetitions are counted (for each attempt)	efficiency of the working memory (VM1), short-term memory (VM2, VM3, VM4, and VM5, VM1–VM5 are successive attempts when patients recall the memorized words during the test), and deferred memory (verbal memory deferred test; VMDT), immediate auditory memory (number of words saved), learning (improvement of results in subsequent repetitions), and deferred memory (remembering repeating words)
GoNoGo test	to press a key when a green square appears on the computer screen (“Go” part) and to refrain from pressing a key when a blue square appears on the screen (“NoGo” part); stimuli are presented in a random manner; the time (ms) of correct “Go” reactions and the number and percentage of correct and incorrect “Go” and “NoGo” reactions are listed	response time under the conditions of the need to control reactions–actions and inhibitions (cognitive control and cognitive inhibition); executive functions
Visuospatial working memory task (VWMT)	to remember the layout of the seven playing cards that were previously presented in different places on the monitor screen	visuospatial memory—correct and incorrect answer and time of reaction

**Table 2 jcm-10-05307-t002:** Demographic and clinical data in all group and in subgroups with and without persistently elevated PSA levels.

Parameter		All *n* = 118	Postoperative PSA < 0.1 ng/mL (*n* = 97)	Postoperative PSA > 0.1 ng/mL (*n* = 21)	d-Cohen	*p*
Age (y)		66.0 (60–70)	66.0 (60.0–70.0)	67.0 (65.0–71.0)	0.82	0.29
BMI (kg/m^2^)		27.2 (25.6–29.7)	26.8 (25.4–29.7)	27.5 (26.0–29.0)	0.16	0.54
Months from surgery (m)		19.0 (13.0–33.0)	19.0 (14.0–33.0)	23.0 (12.0–33.0)	0.03	0.70
Diabetes (*n*, %)		19 (16%)	14 (14.5%)	5 (24%)		0.50
Hypertension (*n*, %)		65 (55%)	59 (61%)	6 (28.5%)		**0.02**
MI (*n*, %)		10 (8.5%)	8 (8%)	2 (9.5%)		0.92
Stroke (*n*, %)		7 (6%)	7 (7%)	0 (0%)		0.60
Education	Basic (*n*, %)	6 (5%)	5 (5%)	1 (5%)		0.31
	Vocational (*n*, %)	33 (28%)	27 (28%)	6 (28.5%)		
	Secondary (*n*, %)	36 (30.5%)	26 (27%)	10 (47.5%)		
	Higher (*n*, %)	43 (36.5%)	39 (40%)	4 (19%)		
Physical activity	None (*n*, %)	40 (34%)	37 (38%)	3 (14%)		0.09
	<1x/week (*n*, %)	24 (20.5%)	17 (17.5%)	7 (33.5%)		
	<3x/week	54 (45.5%)	43 (44.5%)	11 (52.5%)		
GRADE	1 (*n*, %)	70 (59.5%)	63 (65%)	7 (33.5%)		**0.005**
	2 (*n*, %)	35 (29.5%)	27 (28%)	8 (38%)		
	3 (*n*, %)	5 (4%)	5 (5%)	0 (0.0%)		
	4 (*n*, %)	4 (3.5%)	1 (1%)	3 (14.25%)		
	5 (*n*, %)	4 (3.5%)	1 (1%)	3 (14.25%)		
Nicotinism (*n*, %)		53 (45%)	44 (45%)	9 (43%)		0.86

Data are shown as the median (Q25–Q75) or number (%). Inter-group differences were assessed using the Mann–Whitney U test. Effect size was measured using the Cohen d method. BMI—body mass index; MI—myocardial infarction; GRADE: group of grading system classification; y—years; m—months. Significant *p*-values shown in bold.

**Table 3 jcm-10-05307-t003:** R-Spearman correlations of cognitive test results and PSA pre, post, and current.

Parameter	Preoperative PSA	*p*	Post-Surgery PSA	*p*	Current PSA	*p*
SRT_C	0.062	ns.	0.019	ns.	0.016	ns.
SRT_RT	−0.118	ns.	0.043	ns.	0.249	**0.007**
VM_1	0.003	ns.	−0.214	**0.017**	−0.193	**0.036**
VM_2	0.114	ns.	−0.049	ns.	−0.235	**0.01**
VM_3	0.055	ns.	−0.017	ns.	−0.218	**0.017**
VM_4	−0.092	ns.	0.080	ns.	−0.266	**0.003**
VM_5	−0.098	ns.	0.078	ns.	−0.267	**0.003**
VMDT_C	−0.123	ns.	−0.084	ns.	−0.187	**0.047**
GoNoGo_C	0.116	ns.	0.017	ns.	−0.156	ns.
GoNoGo_RT	0.057	ns.	−0.048	ns.	−0.194	**0.035**
GoNoGo IncGO	−0.132	ns.	−0.069	ns.	0.131	ns.
GoNoGo IncNoGo	0.087	ns.	−0.016	ns.	0.108	ns.
VWMT_C	−0.061	ns.	−0.039	ns.	0.131	ns.
VWMT_CRT	0.023	ns.	−0.029	ns.	0.224	**0.014**
VWMT_IRT	0.060	ns.	−0.039	ns.	0.244	**0.007**

SRT_C—simple reaction time test (number of correct answers); SRT_RT—simple reaction time test (average reaction time); VM_1–VM_5—verbal memory (number of words remembered in each of the five attempts); VMDT—verbal memory deferred test (number of words remembered); GoNoGo_C—GoNoGo test (number of correct answers); RT—reaction time; IncGo—number of incorrect Go answers; IncNoGo—incorrect NoGo answers; VWMT_C—visuospatial working memory task (number of correct answers); CRT—average response time for correct answers; IRT—average response time for incorrect answers. PSA—prostate-specific antigen. Significant *p*-values shown in bold.

**Table 4 jcm-10-05307-t004:** R-Spearman correlations of cognitive test results and current PSA level in subgroups with and without diabetes.

Parameter	Current PSA in NONdiabetes Group	*p*	Current PSA in Diabetes Group	*p*
SRT_C	0.044846	ns.	0.003714	ns.
SRT_RT	−0.119912	ns.	0.703496	**0.0007**
VM_1	0.048692	ns.	−0.510703	**0.025**
VM_2	0.089762	ns.	−0.616001	**0.005**
VM_3	−0.030882	ns.	−0.558431	**0.012**
VM_4	−0.066766	ns.	−0.615173	**0.005**
VM_5	−0.009472	ns.	−0.625363	**0.004**
VMDT_C	0.175397	ns.	−0.474987	**0.03**
GoNoGo_C	0.069103	ns.	−0.231986	ns.
GoNoGo_RT	0.117830	ns.	−0.486785	**0.03**
GoNoGo IncGO	−0.054729	ns.	0.311612	ns.
GoNoGo IncNoGo	−0.048036	ns.	0.211985	ns.
VWMT_C	0.175397	ns.	−0.458971	**0.048**
VWMT_CRT	−0.113569	ns.	0.789822	**0.00005**
VWMT_IRT	−0.083593	ns.	0.739536	**0.0002**

SRT_C—simple reaction time test (number of correct answers); SRT_RT—simple reaction time test (average reaction time); VM_1–VM_5—verbal memory (number of words remembered in each of the five attempts); VMDT—verbal memory deferred test (number of words remembered); GoNoGo_C—GoNoGo test (number of correct answers); RT—reaction time; IncGo—number of incorrect Go answers; IncNoGo—incorrect NoGo answers; VWMT_C—visuospatial working memory task (number of correct answers); CRT—average response time for correct answers; IRT—average response time for incorrect answers; PSA—prostate-specific antigen. Significant *p*-values shown in bold.

**Table 5 jcm-10-05307-t005:** R-Spearman correlations of cognitive test and free testosterone in group treated and untreated by hormone therapy.

Parameter	Free Testosterone				Current PSA			
	No Hormone Therapy Group (*n* = 104)	*p*	Hormone Therapy Group (*n* = 14)	*p*	No Hormone Therapy Group (*n* = 14)	*p*	Hormone Therapy Group (*n* = 104)	*p*
SRT_C	0.020066	ns.	0.012728	ns.	0.0184537	ns.	−0.030253	ns.
SRT_RT	−0.183624	0.06	0.167365	ns.	0.261116	ns.	0.001209	ns.
VM_1	−0.216214	**0.02**	0.377769	ns.	−0.236645	ns.	0.080332	ns.
VM_2	−0.166072	ns.	0.311832	ns.	−0.542544	**0.04**	−0.073243	ns.
VM_3	−0.001777	ns.	0.148310	ns.	−0.445146	ns.	−0.126351	ns.
VM_4	0.022967	ns.	0.395777	ns.	−0.336402	ns.	−0.218489	**0.02**
VM_5	0.031529	ns.	0.493676	0.07	−0.191995	ns.	−0.219604	**0.02**
VMDT_C	0.004462	ns.	0.667309	**0.01**	0.077626	ns.	−0.163046	ns.
GoNoGo_C	0.066962	ns.	0.779579	**0.001**	−0.090094	ns.	0.022998	ns.
GoNoGo_RT	−0.082913	ns.	0.485360	0.07	−0.417786	ns.	0.047521	ns.
GoNoGo IncGO	−0.065987	ns.	−0.711890	**0.004**	0.090094	ns.	−0.023911	ns.
GoNoGo IncNoGo	−0.043903	ns.	−0.765654	**0.001**	0.418701	ns.	−0.029561	ns.
VWMT_C	−0.093605	ns.	0.487308	0.07	0.066111	ns.	−0.163046	ns.
VWMT_CRT	0.044880	ns.	−0.393309	ns.	0.681656	**0.007**	0.263424	0.007
VWMT_IRT	0.078058	ns.	−0.178738	ns.	0.739830	**0.002**	0.075388	ns.

SRT_C—simple reaction time test (number of correct answers); SRT_RT—simple reaction time test (average reaction time); VM_1–VM_5—verbal memory (number of words remembered in each of the five attempts); VMDT—verbal memory deferred test (number of words remembered); GoNoGo_C—GoNoGo test (number of correct answers); RT—reaction time; IncGo—number of incorrect Go answers; IncNoGo—incorrect NoGo answers; VWMT_C—visuospatial working memory task (number of correct answers); CRT—average response time for correct answers; IRT—average response time for incorrect answers. Significant *p*-values shown in bold.

**Table 6 jcm-10-05307-t006:** Analysis of the covariance (ANCOVA) of the factors determining the parameters of neuropsychological tests.

	Age		Duration from Surgery		Pre-Treatment PSA		Post-Surgery PSA		Current PSA		Free Testosterone		Total Testosterone		Diabetes		GRADE		Hormone Therapy	
	Wald	*p*	Wald	*p*	Wald	*p*	Wald	*p*	Wald	*p*	Wald	*p*	Wald	*p*	Wald	*p*	Wald	*p*	Wald	*p*
SRT_C	0.8	0.34	26.0	**<0.001**	17.2	**<0.001**	2.7	0.09	16.2	**<0.001**	1.5	0.22	21.8	**<0.001**	2.9	0.08	14.5	**0.002**	3.5	0.06
SRT_RT	0.2	0.61	0.01	0.91	0.002	0.95	0.5	0.47	4.5	0.03	0.36	0.54	9.1	**0.001**	0.03	0.84	1.27	0.73	0.005	0.94
VM_1	3.0	**0.08**	0.05	0.80	1.4	0.23	1.64	0.19	2.08	0.14	2.9	0.08	0.34	0.59	0.56	0.45	0.84	0.83	0.95	0.32
VM_2	0.4	0.52	0.64	0.42	0.05	0.81	0.02	0.87	0.89	s0.34	3.41	0.06	0.03	0.84	5.7	**0.01**	0.95	0.81	0.22	0.64
VM_3	0.4	0.50	0.03	0.86	0.57	0.44	0.91	0.33	1.1	0.29	0.01	0.93	3.22	0.07	8.8	**0.002**	8.2	0.04	0.28	0.59
VM_4	2.9	0.08	5.8	**0.01**	3.4	0.06	0.49	0.48	0.93	0.33	2.2	0.13	0.56	0.45	1.35	0.24	4.1	0.25	0.93	0.33
VM_5	2.9	0.08	5.7	**0.01**	3.4	0.06	0.5	0.48	0.9	0.33	2.24	0.13	0.56	0.45	1.3	0.24	4.1	0.25	0.93	0.33
VMDT_C	19.4	**<0.001**	16.5	**<0.001**	18.3	**<0.001**	0.28	0.59	0.12	0.72	15.8	**<0.001**	0.56	0.45	1.01	0.31	23.3	**<0.001**	7.4	**0.006**
GoNoGo_C	1.22	0.26	1.4	0.22	2.5	0.11	3.8	**0.05**	6.4	0.01	0.14	0.70	12.3	**<0.001**	0.7	0.38	11.4	**0.009**	0.5	0.46
GoNoGo_RT	0.02	0.86	1.64	0.2	2.8	0.09	0.2	0.66	0.24	0.62	0.006	0.93	10.2	**<0.001**	0.54	0.45	2.2	0.52	0.67	0.41
GoNoGo IncNoGo	4.8	0.02	8.5	**0.003**	5.8	**0.01**	0.2	0.6	0.02	0.88	19.9	**<0.001**	1.8	0.17	<0.001	0.95	4.7	0.19	11.1	**<0.001**
VWMT_C	15.2	**<0.001**	12.3	**<0.001**	0.98	0.32	0.53	0.46	0.44	0.50	0.28	0.59	0.97	0.32	3.15	0.07	6.78	0.07	4.2	**0.04**
VWMT_CRT	2.64	0.10	0.37	0.54	1.6	0.20	0.14	0.70	0.23	0.62	5.2	0.02	25.0	**<0.001**	1.8	0.18	8.1	0.04	0.6	0.41

SRT_C—simple reaction time test (number of correct answers); SRT_RT—simple reaction time test (average reaction time); VM_1–VM_5—verbal memory (number of words remembered in each of the five attempts); VMDT—verbal memory deferred test (number of words remembered); GoNoGo_C—GoNoGo test (number of correct answers); RT—reaction time; IncGo—number of incorrect Go answers; IncNoGo—incorrect NoGo answers; VWMT_C—visuospatial working memory task (number of correct answers); CRT—average response time for correct answers. Significant *p*-values shown in bold.

## Data Availability

Data available on request due to ethical policy.

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
