# Peer review of "Prostate-Specific Antigen and Testosterone Levels as Biochemical Indicators of Cognitive Function in Prostate Cancer Survivors and the Role of Diabetes"

_jcm, 2021, doi:10.3390/jcm10225307_

Round 1
Reviewer 1 Report
In this manuscript Popiołek and collaborators investigated the role of prostate-specific antigen (PSA) and testosterone (TT) levels as biochemical indicators of cognitive function in prostate cancer (PCa) patients.
Cognitive function was evaluated by a computer-based test called Neurotest that considers reactions time to stimuli, verbal memory, visual memory...
Authors compared cognitive functions using previous parameters in different clusterized groups of patients, based on PSA values, TT values, Diabetes status.
They concluded that diabetes is associated with the effect of PSA or TT levels on cognitive function, supporting that good quality of care in diabetes patients with PCa is important.
Moreover, PSA levels and free TT levels represent potential biomarkers of cognitive function.
Despite the authors are aware that the small number of subjects included in the study represents a limitations, some additional information would be needed.
Post-operative PSA > 0.2 ng/ml represents a routine parameter used to define biochemical recurrence of PCa after surgery. As suggested by the higher grade of tumors in Table 2 "Postoperative PSA >0,1 ng/ml" Group could be composed by patients with more aggressive and/or advanced diseases.
Do the authors have information on the presence of metastases?
The association between PCa and diabetes is a topic already investigated in past works. The majority of results indicate that diabetes may result in a worse prognosis for men with prostate cancer.
How can the authors exclud that cognitive functions are impaired due to the presence of more advanced tumor disease and the presence of comorbidities with diabetes, and not to PSA levels?
Could PSA level in this specific case simply is an indicator of prostate cancer severity, as is well known?
Author Response
Thank you very much for a thorough reading of our work and for valuable comments and tips
We kindly ask you to analyze the changes described below and answers given for questions based on your review.
- Introduction has been improved.
- The statistical methods have been described.
- Multivariate analysis has been moved from supplementary materials to the main body of the text to make the results more transparent and to show that conclusions are supported by the results
- English language and style have been improved by professional English Editing Services company
Kind regards

Reviewer 2 Report
The biological background to support the aim of the study is poorly described.
The statistical methods are not described .
The statistical evaluations are very poor. The distribution of data of the different variables is not shown. No correlation between the different cognitive tests has been evaluated. The analysis in the subgroups is inappropriate. No multivariate analysis has been performed. I suggest to refer to a statistician to provide robust results.
Some sentences pag 11 lines 194-197 are not appropriate to sintetize the use of PSA in the framework of PC. You can refer to this recent paper:
Ferraro S, et al. Serum prostate specific antigen (PSA) testing for early detection of prostate cancer: Managing the gap between clinical and laboratory practice. Clin Chem 2021;67:602-609
Author Response
Thank you very much for a thorough reading of our work and for valuable comments and tips
We kindly ask you to analyze the changes described below and answers given for questions based on your review.
- Introduction has been supplemented by the biological background to support the aim of the study.
- The statistical methods have been described as suggested.
- Multivariate analysis has been moved from supplementary materials to the main body of the text to make the results more transparent and to show that conclusions are supported by the results
- English language and style have been improved by professional English Editing Services company – you can find attached certificate
Kind regards

Round 2
Reviewer 2 Report
The manuscript has not improved enough and relevant changes require to be introduced.
First no multivariate analysis has been performed yet. Now I strongly recommend to refer to a statistician to provide robust results from this analysis. Only Ancova has been introduced. There are too many tables reporting p values. I strongly suggest to let the tables in the supplementary and comment the main significant results in the Results section.
Some inappropriate sentences are reported on PSA. Page 2 line 68: "PSA is a molecular...". More properly: PSA is the biomarker recommended by clinical practice guidelines for the early detection of prostate cancer (PCa) to rule in patients for prostate biopsy referral, for PCa surveillance and therapeutic monitoring. Indeed PSA is incorporated in the diagnostic algorithms. Look at Ferraro S, et al. Definition of Outcome-Based Prostate-Specific Antigen (PSA) Thresholds for Advanced Prostate Cancer Risk Prediction. Cancers 2021.
The sentence on pag 9 in the Discussion lines 243-249 has been inapropriately reported. It is correct "PSA belongs to the serin protease family and together with other clinical tools(....) is recommended by clinical practice guidelines for the early detection of prostate cancer (PCa) and for therapeutic monitoring [25].
